# Thermoresponsive polymer assemblies via variable temperature liquid-phase transmission electron microscopy and small angle X-ray scattering

Joanna Korpanty [1], Lucas R. Parent [2], Nicholas Hampu[1], Steven Weigand[3] & Nathan C. Gianneschi [1,4,5 ✉]

Herein, phase transitions of a class of thermally-responsive polymers, namely a homo-polymer, diblock, and triblock copolymer, were studied to gain mechanistic insight into nanoscale assembly dynamics via variable temperature liquid-cell transmission electron microscopy (VT-LCTEM) correlated with variable temperature small angle X-ray scattering (VT-SAXS). We study thermoresponsive poly(diethylene glycol methyl ether methacrylate) (PDEGMA)-based block copolymers and mitigate sample damage by screening electron flux and solvent conditions during LCTEM and by evaluating polymer survival via *post-mortem* matrix-assisted laser desorption/ionization imaging mass spectrometry (MALDI-IMS). Our multimodal approach, utilizing VT-LCTEM with MS validation and VT-SAXS, is generalizable across polymeric systems and can be used to directly image solvated nanoscale structures and thermally-induced transitions. Our strategy of correlating VT-SAXS with VT-LCTEM provided direct insight into transient nanoscale intermediates formed during the thermally-triggered morphological transformation of a PDEGMA-based triblock. Notably, we observed the temperature-triggered formation and slow relaxation of core-shell particles with complex microphase separation in the core by both VT-SAXS and VT-LCTEM.

[1] Department of Chemistry, International Institute for Nanotechnology, Chemistry of Life Processes Institute, Simpson Querrey Institute, Northwestern University, Evanston, IL 60208, USA. [2] Innovation Partnership Building, University of Connecticut, Storrs, CT 06269, USA. [3] DuPont−Northwestern−Dow Collaborative Access Team (DND-CAT) Synchrotron Research Center, Northwestern University, Argonne, IL 60208, USA. [4] Department of Materials Science & Engineering, Northwestern University, Evanston, IL 60208, USA. [5] Department of Biomedical Engineering and Department of Pharmacology, Northwestern University, Evanston, IL 60208, USA. ✉email: nathan.gianneschi@northwestern.edu

Thermoresponsive polymers are used in numerous technological applications, including biomedicine, insulator materials, and tissue engineering[1–5]. The most widely studied thermoresponsive polymers exhibit a lower critical solution temperature (LCST) in water. Upon heating above the LCST, such polymers undergo an entropically driven phase separation that coincides with a coil-to-globule transformation[6–8]. The resulting phase transition induces morphological and rheological changes, attracting attention as so-called "smart" materials for stimuli-responsive drug carriers[9], nanoreactors[10], and industrial coatings[11].

Despite the ubiquity of thermoresponsive polymeric materials, currently, we lack well-established, direct techniques for elucidating their elevated temperature, solution-phase, nanoscale morphologies, and dynamics. Presently, the accepted workflow for analyzing solvated thermoresponsive soft nanomaterials at elevated temperatures consists of scattering techniques in combination with static imaging via electron microscopy. Scattering techniques, including variable temperature dynamic light scattering (VT-DLS) and variable temperature small angle X-ray scattering (VT-SAXS), require prior assumptions about the system so that raw data can be fit to models, often creating challenges in assigning nanostructure morphologies[12–14]. Further, as bulk averaging techniques, these indirect methods provide limited information about discrete nanostructures[15]. Alternatively, direct imaging by traditional TEM methods at temperature is typically not feasible for nanomaterials that can undergo thermally reversible transitions. Specifically, dry state TEM produces drying and cooling artifacts not reflective of native sample morphology[16,17]. Conversely, cryogenic (cryo) TEM samples can in principle be prepared at temperature within humidity-controlled environments before vitrification[18,19], but in actuality, the temperature of microliter droplets cannot be precisely controlled. Moreover, blotting can generate artifacts, including shear stresses and particle packing[20,21]. Above all, these traditional microscopy methods only provide static snapshots to guide the interpretation of in situ scattering studies.

Herein, thermoresponsive polymeric materials are examined by liquid-cell transmission electron microscopy (LCTEM), a nascent technique for imaging solvated nanomaterials and their dynamics. With the advent of variable temperature (VT) LCTEM[22], which allows for in situ heating, the method can potentially provide unparalleled insight into thermally responsive systems. The typical experimental setup for LCTEM employs two silicon microchips with electron transparent silicon nitride (SiN$_x$) windows (Supplementary Fig. 1). A liquid sample is sealed between the two chips, allowing for analysis in an electron microscope[23]. Thus, VT-LCTEM should enable visualization of nanostructures at elevated temperatures, including those formed from LCST-type polymers, providing mechanistic insight into the thermal transition behavior. Here, we examine three such phase transitions for homopolymers, diblocks, and triblock copolymers (Fig. 1).

First, we optimize VT-LCTEM conditions to elucidate the LCST-transition of poly(diethylene glycol methyl ether

methacrylate) (PDEGMA, Fig. 1). In studying PDEGMA, we directly observe elevated temperature nanostructures in a solvated LCST-type polymer for the first time and gain insight into the LCST-transition dynamics. We next apply VT-LCTEM to directly observe the formation of elevated temperature nano-assemblies in PDEGMA-b-poly(ethylene glycol) (PEG). Here, PEG serves as a non-responsive hydrophilic block, and the change in hydrophilicity of PDEGMA upon heating prompts assembly (Fig. 1). Finally, we study PEG-b-PDEGMA-b-poly(2-hydroxypropyl methacrylate) (PHPMA) to observe the temperature-triggered morphological transformation of pre-formed nanostructures. Here, PHPMA is hydrophobic, making the polymer amphiphilic at room temperature. The PDEGMA LCST-transition alters the hydrophobic-to-hydrophilic balance of the triblock, triggering a morphological transformation (Fig. 1). Given its complexity, we apply VT-SAXS to analyze the transformation of the triblock and rigorously define the elevated temperature structure.

## Results and discussion

**VT-LCTEM study of PDEGMA.** First, we synthesized PDEGMA (Fig. 2). We measured a cloud point of 45 °C via VT-DLS that remained stable upon further heating (Fig. 2a, b, Supplementary Fig. 2). For PDEGMA and aqueous organic systems generally, a critical consideration for LCTEM is careful electron flux (e$^-$Å$^{-2}$s$^{-1}$) selection[24,25]. While a higher flux increases signal-to-noise and improves contrast, beam-induced phenomena often become significant with increasing instantaneous flux and cumulative fluence (e$^-$Å$^{-2}$).

To minimize such artifacts, we screened flux, fluence, and solvent conditions over separate LCTEM experiments and evaluated PDEGMA survival via *post-mortem* matrix-assisted laser desorption/ionization imaging mass spectrometry (MALDI-IMS, Supplementary Movies 1-n11)[26–28]. We chose to screen hydroxyl radical (•OH) scavengers, like isopropanol (IPA)[28], t-butanol (t-BuOH)[29], and dimethyl sulfoxide (DMSO)[30], as •OH is generally the most destructive radiolysis product towards polymers[31–33]. Additionally, we tested deuterated[34] and degassed[35] water, which are hypothesized to mitigate radiolytic damage compared to water. *Post-mortem* MALDI-IMS indicated that PDEGMA survived at higher fluxes and fluences in 5% t-BuOH, 5% IPA, and pure D$_2$O (Fig. 2c, Supplementary Figs. S3, S4).

As PDEGMA also survived LCTEM imaging under purely aqueous conditions at a flux of 0.8 e$^-$Å$^{-2}$s$^{-1}$ and a low fluence (< 10 e$^-$Å$^{-2}$), we employed these conditions with stroboscopic imaging[22,36–39] to probe the purely aqueous LCST-transition (Fig. 2d–f). To ensure the LCST would occur in the diffusion-limited, confined liquid-cell, we used 20 mg mL$^{-1}$ PDEGMA and heated it to 60 °C. With heating, we visualized polymer phase separation, demonstrating the ability of VT-LCTEM to directly image a solvated LCST-type polymer at temperature (Fig. 2f). Cooling did not completely redissolve the sample during the 60 min it was monitored, likely due to diffusion constraints and

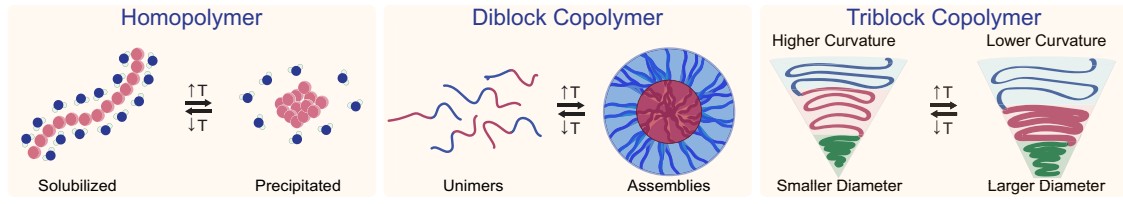

**Fig. 1 Thermal transitions in LCST-type polymers.** Thermally triggered transformations in an LCST-type polymer for a homopolymer undergoing precipitation (left), a diblock copolymer undergoing assembly (center), and a triblock copolymer undergoing a morphological transformation (right). Note that these transformations may show partial reversibility.

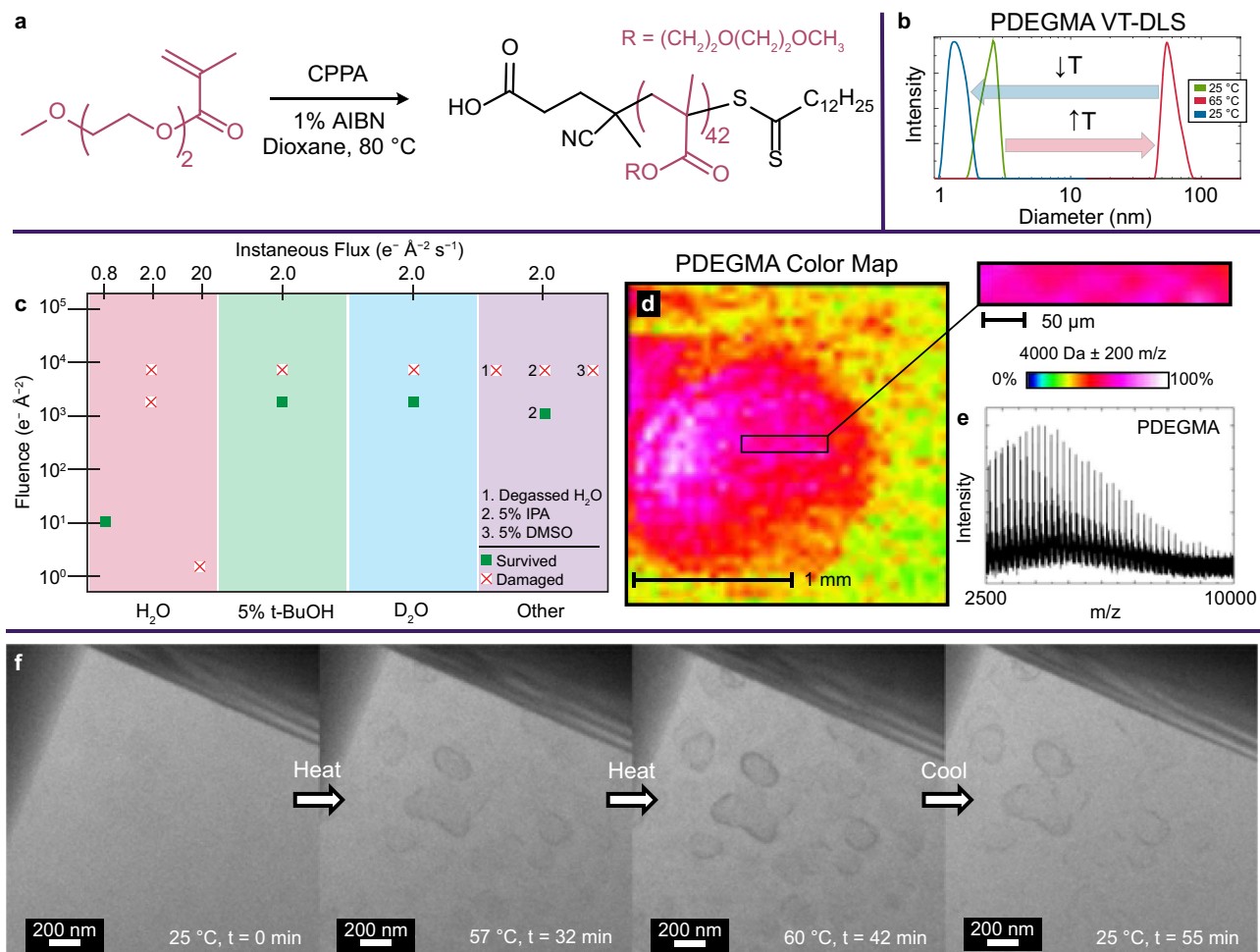

**Fig. 2 VT-LCTEM and MALDI-IMS analysis of PDEGMA homopolymer. a** Polymerization of DEGMA to form a PDEGMA homopolymer, where CPPA denotes 4-cyano-4-(phenylcarbonothioylthio)pentanoic acid and AIBN denotes azobisisobutyronitrile. **b** VT-DLS of 5 mg mL$^{-1}$ PDEGMA in water heated from 25 °C (green) to 65 °C (red) and cooled to 25 °C (blue). **f** Single frames of VT-LCTEM experiment on 20 mg mL$^{-1}$ PDEGMA in water imaged at a flux of 0.8 e$^-$Å$^{-2}$s$^{-1}$ and heated to 60 °C. **c** Plot showing survival (green square) or destruction (white square with red X) of PDEGMA homopolymer under different imaging and solvent conditions, as measured by MALDI-IMS. The screened additives were isopropanol (IPA), t-butanol (t-BuOH), dimethyl sulfoxide (DMSO), and deuterium oxide (D$_2$O). **d** MALDI-IMS colormap of bottom chip with inset showing imaged window from VT-LCTEM experiment on PDEGMA with a mass filter of 4000 ± 200 m/z displayed as 0–100% of total intensity on a logarithmic scale. **e** Mass spectrum of imaged region. Source data are provided as a Source data file.

the propensity for polymers to stick to SiN$_x$, which occurred even in unimaged controls (Supplementary Fig. 5).

**VT-LCTEM study of PEG-*b*-PDEGMA**. Having optimized imaging conditions to study PDEGMA, we sought to visualize assembly of aqueous PEG-*b*-PDEGMA (Fig. 3). The formation of nano-assemblies upon heating was confirmed via VT-DLS (Fig. 3a, b). Using a flux of 0.8 e$^-$Å$^{-2}$s$^{-1}$, the same conditions where PDEGMA survived, PEG-*b*-PDEGMA assemblies formed at 50 °C from an aqueous solution without pre-formed structures (Fig. 3c, d). The predominant morphology by VT-LCTEM was small (30-50 nm) micellar nanoparticles with several larger (~ 200 nm) vesicular structures.

We measured the average background intensity for several regions surrounding the assemblies and found the intensity increased upon heating above the LCST and decreased upon cooling (Supplementary Fig. 6). This change in background intensity, which reflects a change in mass-thickness, was likely caused by liquid exclusion upon heating, as the PDEGMA became more hydrophobic and expelled surrounding solvent, followed by local rehydration upon cooling.

With the PEG-*b*-PDEGMA system, we demonstrate the first example of a thermoresponsive diblock copolymer assembly being directly observed in the solution. Though PEG-*b*-PDEGMA was too large to ionize by MALDI-IMS, PDEGMA survival under the same imaging conditions suggests the diblock likewise survives. Moreover, VT-LCTEM control experiments showed similar assemblies formed when the sample was left unimaged until reaching the LCST, indicating observed assemblies were thermally driven, not beam-induced (Supplementary Fig. 5). Given the persistence of the LCST in alcohol mixtures[40] and the ability of alcohols to scavenge destructive •OH radicals, we conducted VT-LCTEM of PEG-*b*-PDEGMA in 15% IPA in water and observed the formation of vesicular assemblies at 65 °C (Fig. 3e and Supplementary Fig. 7).

**VT-LCTEM study of PEG-*b*-PDEGMA-*b*-PHPMA**. Next, we sought to study a more complex triblock system, where the polymer is amphiphilic at room temperature with the addition of a non-responsive hydrophobic block (Fig. 4). Heating prompts a morphological transformation, as PDEGMA alters the hydrophilic-to-hydrophobic balance of the polymer (Fig. 4a). At

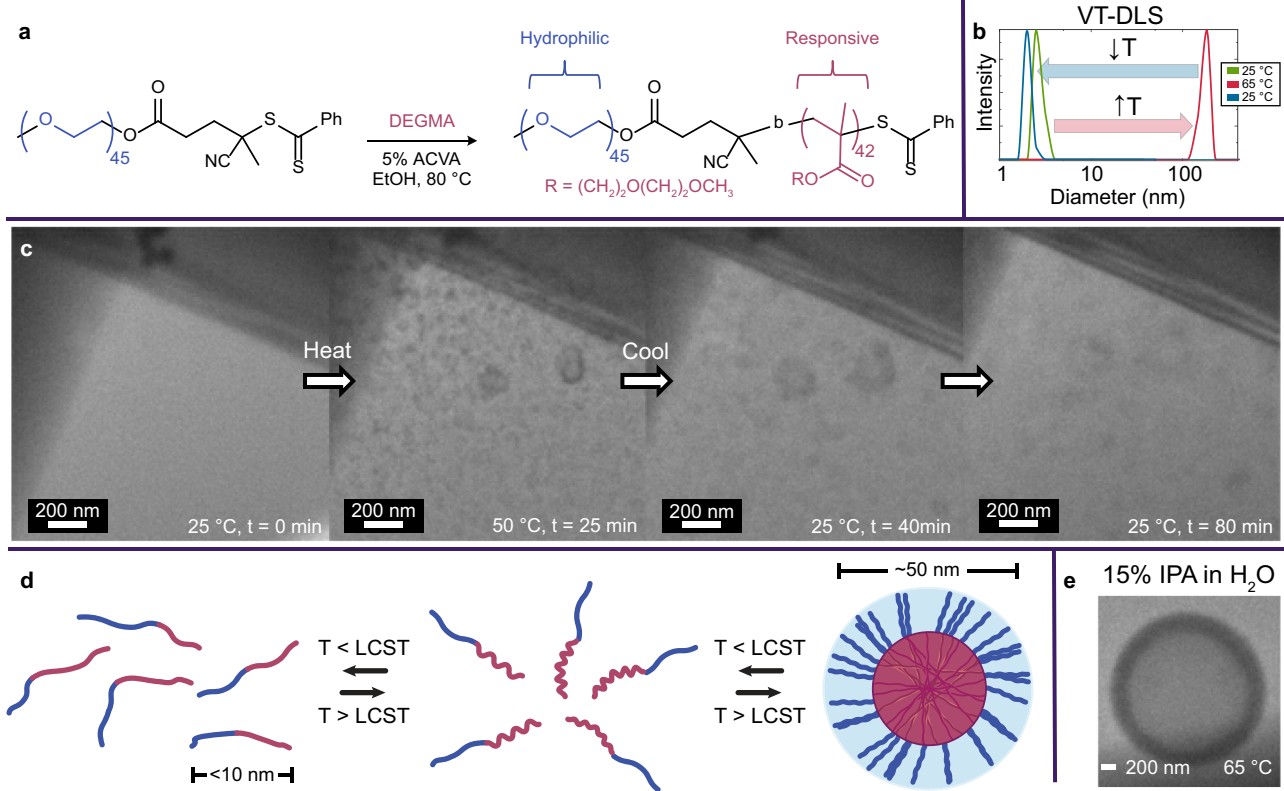

**Fig. 3 A temperature-responsive diblock copolymer imaged via VT-LCTEM. a** Chain extension of 2 kDa PEG macro-CTA with DEGMA to form a diblock copolymer, where ACVA denotes 4,4′-azobis(4-cyanovaleric acid). **b** VT-DLS of 5 mg mL$^{-1}$ PEG-*b*-PDEGMA in water heated from 25 °C (green) to 65 °C (red) and cooled to 25 °C (blue). **c** Single frames of VT-LCTEM experiment on 10 mg mL$^{-1}$ PEG-*b*-PDEGMA in water imaged at a flux of 0.8 e$^-$Å$^{-2}$s$^{-1}$ and heated to 60 °C. **d** Schematic of hydrophilic homopolymer chain-extension to yield a temperature-responsive diblock copolymer that reversibly undergoes assembly upon heating above the LCST. **e** Formation of vesicular assemblies at 65 °C during a VT-LCTEM experiment on PEG-*b*-PDEGMA. Source data are provided as a Source data file.

room temperature, the dry state TEM of the triblock solution showed micellar nanoparticles ~ 60 nm in diameter (Fig. 4b). VT-DLS indicated a transformation from small assemblies (30–60 nm) into larger structures (200–300 nm) upon heating (Fig. 4c).

We directly imaged a concentrated solution (15 wt%) of thermoresponsive PEG-*b*-PDEGMA-*b*-PHPMA via VT-LCTEM at the same imaging conditions where PDEGMA survived (Fig. 3d, e, Supplementary Fig. 8). Using a flux of 0.8 e$^-$Å$^{-2}$s$^{-1}$, at room temperature, we observed small micelles, ~ 60 nm, which we analyzed using image processing (Fig. 4d, Supplementary Fig. 9). Heating the sample prompted the formation of larger, higher contrast intermediates at t = 32 min, because of the LCST-transition of the PDEGMA block. The transition drove PDEGMA into the particle core with PHPMA (Fig. 4d). The intermediates that formed immediately after heating above the LCST were associated with a significantly increased background liquid intensity, indicating a decrease in the surrounding liquid thickness, which could have been caused by water exclusion from the concentrated assemblies (Supplementary Fig. 6).

Continuing to maintain a temperature of 60 °C, the formation of high contrast intermediates was followed by the appearance of a more dispersed halo surrounding the dense core and a return of the initial liquid thickness at t = 40 min (Supplementary Fig. 6). The clear core-shell structure observed at t = 40–50 min likely reflected exclusion of PHPMA from the core by PDEGMA, causing an increased shell contrast, as PHPMA is hydrophobic. Exclusion of PHPMA from the core was likely caused by enthalpically unfavorable interfacial contacts between PHPMA

and PDEGMA upon the PDEGMA LCST-transition. It is also possible that the intermediates observed at t = 40–50 min reflect incomplete packing of PDEGMA into the core, leaving the PDEGMA block partially packed into the dense core and partially exposed to the surrounding water.

Upon cooling, the initial morphology was not restored, as observed by both VT-DLS and VT-LCTEM (Fig. 4d, Supplementary Fig. 10). It is likely that the PHPMA vitrified before the PDEGMA could reswell, due to the glass transition temperature ($T_g$) of hydrated PHPMA, at 47 °C[41], being below the LCST of PDEGMA, ~ 50 °C. The observed liquid thinning in the VT-LCTEM experiment may have resulted from PDEGMA excluding surrounding water, an inherent component of the LCST-transition (Supplementary Fig. 6)[8]. Since the LCST-transition is a manifestation of polymer–solvent interactions, this observation exemplifies the unique ability of LCTEM to probe the subtleties of polymer–solvent interactions.

To further analyze LCTEM data, we utilized image processing (Fig. 4e, f). Image processing highlighted the initial presence of small particles, which likely aggregated upon the onset of the PDEGMA LCST (Fig. 4d, e, first frame). This aggregation led to the formation of a larger, more hydrophobic core (Fig. 4e, second and third frames). As the PDEGMA block expelled more water during its LCST-transition, it likely excluded PHPMA from the core to minimize unfavorable interactions, yielding an intermediate contrast shell (Fig. 4d, fourth frame).

The intermediate contrast of the shell was better captured using false coloring, whereby each pixel value range was assigned a red, green, or blue value. Applying a 6-shade coloring

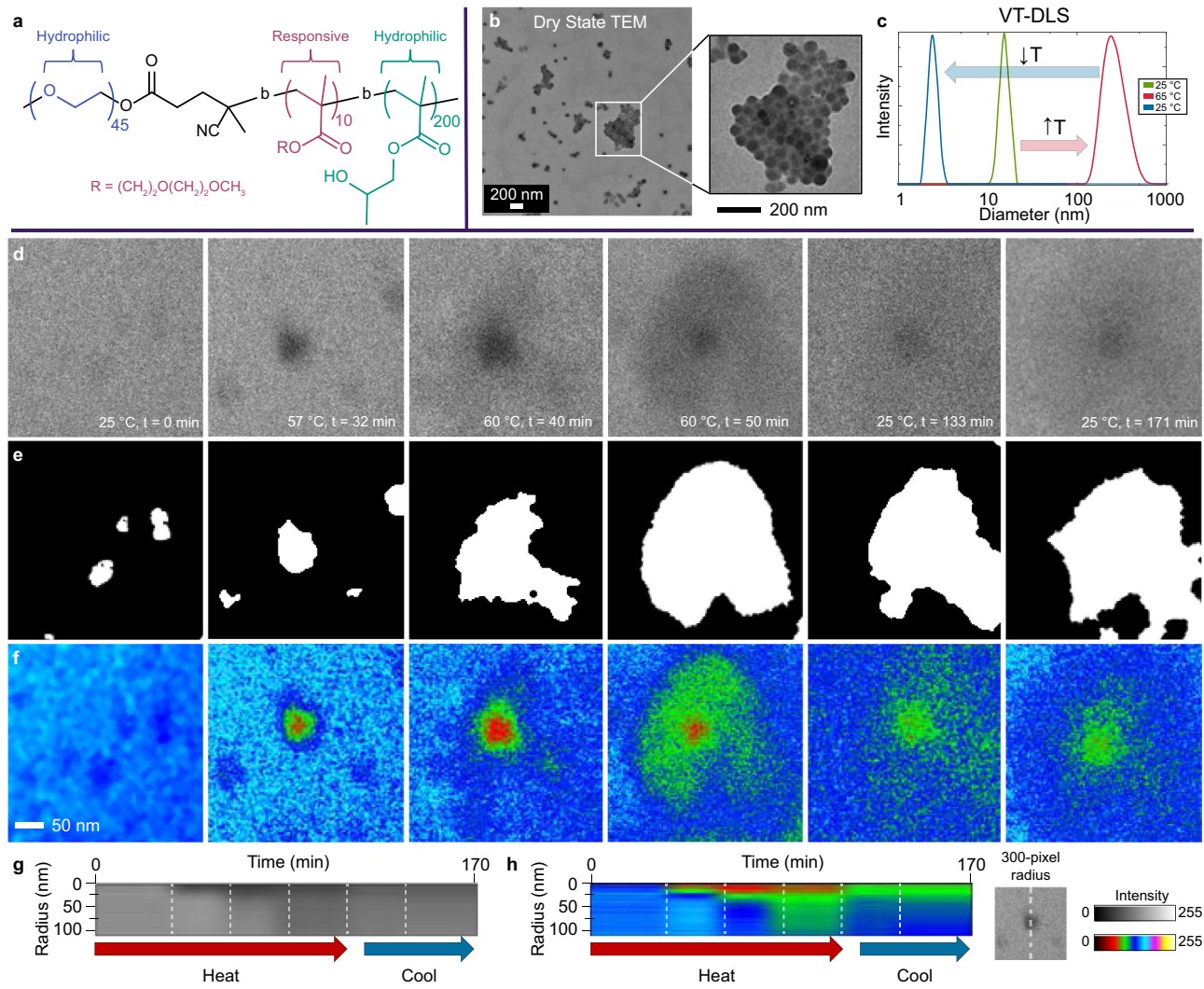

**Fig. 4 VT-LCTEM analysis of PEG-*b*-PDEGMA-*b*-PHPMA triblock copolymers. a** Structure of PEG-*b*-PDEGMA-*b*-PHPMA triblock copolymer used for LCTEM studies. **b** Dry state TEM image of the triblock copolymer, highlighting formation of 60 nm micelles. **c** VT-DLS of 5 wt% triblock in water heated from 25 °C (green) to 65 °C (red) and cooled to 25 °C (blue). **d** Single frames of raw, cropped regions of interest for each timepoint are shown in Supplementary Fig. 8. These images are from a VT-LCTEM experiment on 15 wt% triblock in water imaged at a flux of 0.8 e−Å−2s−1 and heated to 60 °C. **e** Segmented region of interest for each timepoint. **f** Each timepoint with a 6-shade false color filter. **g** Grayscale radial profile of raw images with a 300-pixel radius at the dense center of each nanostructure. **h** 6-shade false color radial profile of raw images with a 300-pixel radius at the center of each nanostructure. Source data are provided as a Source data file.

highlighted the formation of the disperse shell surrounding the high contrast core (Fig. 4f, fourth frame). Cooling the sample then led to reduced density of the core and shell (Fig. 4e, f, fifth and sixth frames). Despite what appeared as full re-solvation of the corona, the core appeared trapped, and the original morphology was not restored over the time monitored. That is, we did not observe fission into small particles. In addition to image processing, we measured the radial profile centered on the particle core (Fig. 4g, h). This profile highlighted the increased contrast of the core and shell upon heating with an increase in contrast from PDEGMA and exclusion of PHPMA from the core. Cooling then led to decreased contrast in the core and even more so in the shell.

**VT-SAXS study of PEG-*b*-PDEGMA-*b*-PHPMA.** To gain a deeper understanding of the transformations directly observed by VT-LCTEM, we utilized VT-SAXS (Fig. 5). SAXS has been used extensively to study thermal transitions of soft matter[42,43], polymerization induced self-assembly[44], lipoprotein phase transitions[45,46], and responsive hydrogels[47]. Interpretation of SAXS data relies upon fitting shape-dependent models to experimental data to understand nanostructure morphology. However, selecting the appropriate model for data fitting is non-trivial, and commonly, TEM data inform model selection. Even for previous studies that employed direct imaging to guide SAXS interpretation[44–46], only static snapshots were obtained and mechanisms for transformations of individual nanostructures were inferred. Accordingly, here we employed VT-SAXS (Fig. 5a) with VT-LCTEM to gain insight into the dynamics of individual nanostructures through both in situ imaging and scattering.

With the initial hypothesis that the triblock forms spherical micelles with a PHPMA core, a PDEGMA shell, and a PEG corona below the LCST of PDEGMA, we fit the experimental scattering data acquired at 30 °C to a core-shell-shell sphere form factor. The fitted curve captured the experimental data reasonably well for intermediate and high *q*, although significant deviation was observed in the low *q* region, which we attribute to the presence of loose micellar aggregates (Fig. 5b). The fit yielded a core radius of 18 nm, a

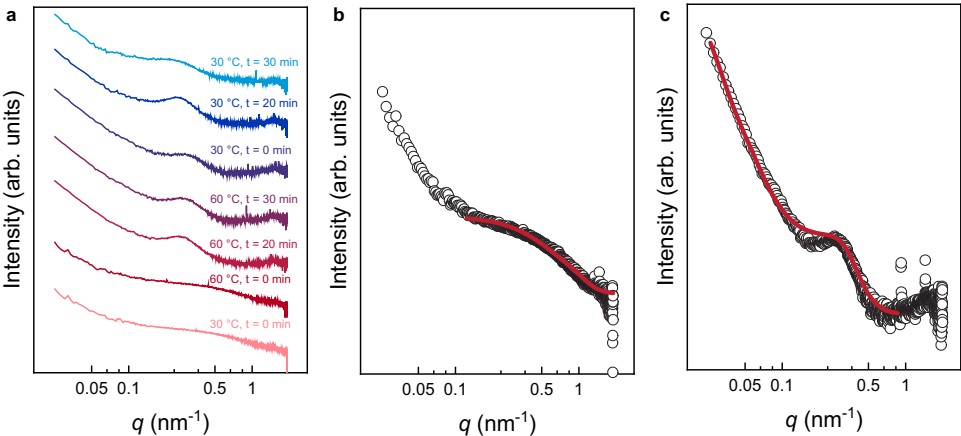

**Fig. 5 SAXS scattering data for PEG-*b*-PDEGMA-*b*-PHPMA triblock copolymer measured at 1 wt%. a** Background subtracted temperature sweep SAXS data with individual traces offset vertically by an arbitrary factor. **b** Scattering data with core-shell-shell sphere form factor fit for sample at 30 °C before heating. **c** Scattering data with sphere and Lorentzian form factor fit for sample held at 60 °C for 30 min. Source data are provided as a Source data file.

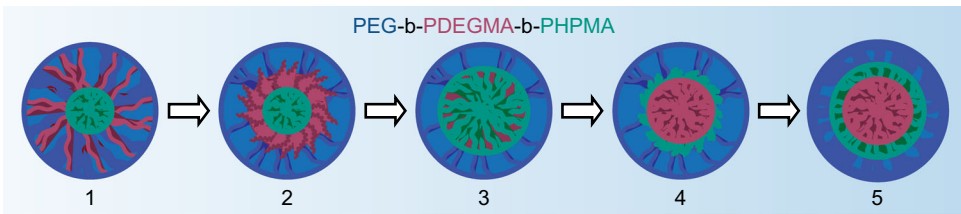

**Fig. 6 Proposed mechanism for temperature-triggered morphological transformation in triblock.** Blue: polyethylene glycol (PEG). Red: poly(diethylene glycol methyl ether methacrylate) (PDEGMA). Green: poly(2-hydroxypropyl methacrylate) (PHPMA).

PDEGMA inner shell thickness of 1.5 nm, and a PEG outer shell thickness of 4.2 nm. These sizes are reasonably consistent with the DLS and TEM measurements, supporting the robustness of our fitting process and demonstrating the potential for using SAXS as a complementary technique to LCTEM (Table S1).

Heating the sample to 60 °C resulted in the formation of a broad peak in the high $q$ region of the SAXS data, which supports the LCTEM observation that a morphological change occurs above the PDEGMA LCST. Based on the low degree of polymerization of the PDEGMA block compared to the PHPMA block, we speculated that the micelle core would restructure above the LCST to accommodate PDEGMA in the core interior and relieve the large entropic penalty for arranging the shorter PDEGMA block on the convex side of the core-shell interface. The LCST of PDEGMA is higher than the $T_g$ of PHPMA, allowing the mobile PHPMA core to accommodate the collapsed PDEGMA domain at 60 °C. Furthermore, the persistence of the broad peak even after cooling back to 30 °C suggests that the PDEGMA domain is trapped within a vitrified PHPMA matrix and is unable to rehydrate below the LCST. Using this hypothesized PDEGMA core, PHPMA inner shell, and PEG outer shell configuration as our initial model, we attempted to fit the 60 °C SAXS data to the same core-shell-shell sphere form factor that we successfully used for the 30 °C data. However, a satisfactory fit was unobtainable using physically relevant fitting parameters. To confirm that the fitting challenge was unrelated to our choice of the initial model, we also attempted to fit the data to the core-shell-shell form factor assuming a PHPMA core, a PDEGMA inner shell, and a PEG outer shell. Again, a satisfactory fit was unobtainable. Therefore, we speculated that the true micelle structure was more complex than the concentric spheres assumed by the core-shell-shell model. Instead, we hypothesized that the PHPMA and PDEGMA domains might be microphase segregated within the micelle core. To test this hypothesis, we then fit the experimental scattering data obtained at 60 °C to the sum of a spherical and broad peak form factor, where the

spherical form factor captures the overall micelle shape, and the broad peak form factor captures the phase segregated core (Fig. 5c). This choice of scattering model yielded an acceptable fit to the experimental data with a micelle radius of 95 nm and a 30 nm spacing between scattering inhomogeneities within the micelle core. The fitted values for the micelle radius correspond well to those measured by LCTEM and VT-DLS, while the 30 nm length scale is reasonably consistent with the size of the high contrast region observed in the micelle interior by LCTEM.

By VT-SAXS, the triblock did not fully relax to its original structure even 30 min after cooling (Fig. 5a). The slow relaxation suggests that enthalpically unfavorable mixing of PDEGMA and PHPMA in the core upon heating led to the exclusion of PHPMA from the inner core so that isolated PDEGMA domains were surrounded by a PHPMA shell (Fig. 6). This result, which was ambiguous by VT-LCTEM in isolation, shows the value of coupling VT-LCTEM with VT-SAXS, as the combination of the two techniques allowed us to propose a mechanism for the triblock thermal transformation (Fig. 6). These results highlight a new potential workflow for characterizing stimuli-responsive soft materials, where insights from SAXS can refine and clarify ambiguous LCTEM features and vice versa, offering a powerful set of new strategies to enhance our current understanding of complex soft nanomaterials.

In summary, we have gained insight into complex polymeric nanostructures and their dynamics by studying thermoresponsive poly(diethylene glycol methyl ether methacrylate) (PDEGMA)-based polymers. We established general guidelines for observing thermoresponsive polymeric materials and processes in situ. Notably, we highlight the importance of optimizing LCTEM conditions coupled with *post-mortem* analysis. Particularly, for the PDEGMA triblock system, optimized VT-LCTEM conditions enabled us to gain direct mechanistic insight and observe intermediate structures formed during the thermally triggered

morphological transformation of the polymer. Correlating VT-LCTEM with VT-SAXS for the triblock provided key insights into an intricate phase transition, showing the value of leveraging the techniques in tandem. Our general workflow shows the potential of LCTEM, coupled with SAXS, to answer fundamental questions about functional and responsive nanomaterials, and our approach can be extended to study nanoscale processes of high scientific importance, such as drug encapsulation and release[9,17,48] from nanocarriers that may occur in a phase-dependent fashion, influencing carrier design and synthesis.

## Methods

**General information**. All materials were purchased from Sigma or TCI chemicals. All monomers were filtered through basic alumina to remove inhibitors, and all other materials were used as received unless otherwise noted. The synthetic details for polymeric materials are provided in the supplementary information.

**LCTEM imaging**. The Protochips Poseidon Select Heating holder was used to collect LCTEM data. Milli-Q water was used to prefill the lines of the holder in all LCTEM experiments. LCTEM chips with 50-nm-thick, 200 μm × 50 μm window $SiN_x$ membranes were cleaned in acetone followed by methanol, dried, and subsequently glow discharged in a PELCO easiGlow glow discharge unit for 5 min. Next, 0.5 μL of the sample was pipetted manually onto the bottom chip, and then the liquid-cell was assembled with the windows (50 μm × 200 μm) aligned perpendicularly (50 μm × 50 μm LCTEM viewing area), and the lines of the holder were sealed off without external flow. In situ liquid flow using the inlet/outlet microfluidic lines of the Poseidon holder was used in several experiments (Supplementary Figs. 17, 18).

Experiments were performed using a JEM-ARM300F (JEOL Ltd., Tokyo, Japan) operated at 300 keV and a JEM-ARM200CF (JEOL, Ltd., Tokyo, Japan) operated at 200 keV. Micrographs were recorded on a 2k × 2k Gatan OneView-IS CCD camera (Gatan Inc., Pleasanton, CA, USA) using Gatan Digital Micrograph image acquisition software (Roper Technologies, Sarasota, FL). The electron flux values used in LCTEM experiments were calculated using the beam current for each aperture selection, as measured by a Faraday Holder through a vacuum, and the beam diameter incident upon the sample. Two experiments performed on this microscope made use of the K3-IS direct electron detector, and for these experiments, the electron dose was directly measured by the detector (Supplementary Fig. 7c, Supplementary Fig. 15). Immediately following LCTEM experiments, the $SiN_x$ chips were carefully separated and allowed to dry.

**Image processing**. We performed image processing using the software Fiji (Fig. 5). First, we cropped a region of interest in a fixed area for each timepoint (Fig. 5a). We binned the cropped images (2 × 2 average), applied a gaussian filter ($\sigma = 1$), thresholded each cropped region of interest, and subtracted features <5 pixels$^2$ from each image (Fig. 5b).

**MALDI-IMS**. LCTEM chips, with their $SiN_x$ membranes facing upwards were adhered to the conductive face of an ITO-coated glass slide with 70–100 ohms resistivity (Bruker Daltonics), using ~0.5 μL nail polish and allowed to dry. To equalize the height difference from $SiN_x$ chips on the slide (~0.25 mm), four pieces of Scotch tape were applied to both short edges of the slide on the same side. As a control, unimaged PDEGMA in deionized water was drop casted onto a clean liquid-cell chip. All chips were coated with trans-2-[3-(4-tert-butylphenyl)-2-methyl-2-propenylidene]mal-ononitrile (DCTB) matrix in (20 mg mL$^{-1}$ in acetonitrile).

Slides were mounted into an MTP Slide Adapter II and loaded onto a Bruker Rapiflex MALDI-ToF mass spectrometer for analysis using the flexControl software (Bruker Daltonics). Samples were analyzed by MALDI-MS under either reflector positive mode (1400–10000 Da) using a 355 nm smartbeam 3D laser with a 50 μm focus diameter and 200 Hz frequency, a constant laser power of 75%, and a sum of 500 shots per spectrum. Spectra were collected using an accelerating voltage of 20 kV and detector gain of 792 V. Region of interest (ROI) mapping was performed at a raster width of 50 μm, and image analysis was performed in flexImaging software (Bruker Daltonics).

**VT-SAXS**. SAXS experiments were conducted at the 5-ID-D beamline of the Dupont-Northwestern-Dow Collaborative Access Team (DND-CAT) at the Advanced Photon Source, Argonne National Laboratory. Samples of 1% w/w polymer in water were prepared and loaded into 1.5 mm quartz capillaries. The capillaries were then sealed with epoxy to prevent solvent evaporation prior to data acquisition. Capillaries were loaded into a multicapillary holder and scattering patterns were first acquired at room temperature. Samples were then heated to 60 °C at a rate of 1 °C/min. After reaching the temperature set point, patterns were obtained every 2 min for 30 min. Samples were then cooled to 30 °C at 1 °C/min. After reaching the set point, patterns were acquired every 2 min for 30 min.

Two-dimensional scattering patterns were obtained from 10 s exposure using a Rayonix MX170-HS CCD area detector using a 0.5 s exposure time to X-rays with a wavelength of $\lambda = 0.7293$ Å and a sample-to-detector distance of 8.5 m. The 2D data were azimuthally averaged to yield 1D scattering patterns as intensity versus $q$. Incoherent background scattering was measured by acquiring scattering patterns for a water-loaded capillary in the absence of polymer. The solvent data was fit to a power law of the form $I(q) = A + Bq^{-m} + Cq^2$ and subtracted from the polymer data. Select scattering patterns were fit to a model comprising the sum of a spherical form factor and a broad Lorentzian peak. The Lorentzian peak was required to obtain a satisfactory fit at high $q$.

## Data availability.

We declare that all other data supporting the findings of this study are available within the article and Supplementary Information files and are also available from the corresponding author upon request. Source data files are available for the data shown in Figs. 2b, 2e, 3b, 4c, 5a, 5b, and 5c. Source data are provided with this paper.

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

## Acknowledgements

Research in the N.C.G. group was conducted with government support under and awarded by DoD through the ARO (W911NF-17-1-0326) and a MURI grant (W911NF-15-1-0568). In addition, the authors thank the NSF for support of both the L.R.P and N.C.G. groups through a joint research grant (CHE-MSN 1905270). This research used EPIC facility of Northwestern University's NUANCE Center, which has received support from the Soft and Hybrid Nanotechnology Experimental (SHyNE) Resource (NSF ECCS-1542205), the MRSEC program (NSF DMR1720139) at the Materials Research Center; the International Institute for Nanotechnology (IIN), the Keck Foundation, and the State of Illinois, through the IIN. This work also made use of the IMSERC at Northwestern University, which has received support from the Soft and Hybrid Nanotechnology Experimental (SHyNE) Resource (NSF ECCS-1542205); the State of Illinois and International Institute for Nanotechnology (IIN). Research reported in this publication was supported in part by instrumentation provided by the Office of The Director, National Institutes of Health of the National Institutes of Health under Award Number S10OD026871. The content is solely the responsibility of the authors and does not necessarily represent the official views of the National Institutes of Health. Use of the Advanced Photon Source (APS) at Argonne National Laboratory was supported by the U.S. Department of Energy, Office of Science, under Contract DE-AC02-06CH11357. SAXS measurements were performed at the DuPont-Northwestern-Dow Collaborative Access Team (DND-CAT) located at Sector 5 of the APS, supported by E. I. DuPont de Nemours and Co., The Dow Chemical Company, and Northwestern University. J.K. gratefully acknowledges support from the Ryan Fellowship and the International Institute for Nanotechnology at Northwestern University.

## Author contributions

J.K., L.R.P. and N.C.G. devised the project. J.K. performed LCTEM, MALDI-IMS, synthesized all materials, and drafted the paper. L.R.P. performed several LCTEM experiments. S.W. performed SAXS measurements and N.H. aided in SAXS data analysis. All authors wrote and edited the manuscript. All authors have given approval to the final version of the manuscript.

## Competing interests

The authors declare no competing interests.
