## [Peer Review File · Nature Communications]

REVIEWER COMMENTS

Reviewer #1 (Remarks to the Author):

This is an interesting study to visualize stimulus-responsive behavior of thermo-responsive polymers by LC-TEM. Overall, there are two major concerns related to the chemistry of the homopolymers and copolymers, and TEM imaging. The authors should address these critical comments:

- The amount of initiator (AIBN/ACVA) seems to be very high (up to 10%) for a controlled/living radical polymerization (RAFT). Usually, a less than 0.1mol% is used for living/controlled radical polymerization. Using high quantity of initiator would decrease the livingness of the polymerization and increase polydispersity. The authors should show data indicate the rational behind this system choice.
- The kinetics of the RAFT polymerizations and the corresponding polydispersity should also be shown. This can demonstrate the linearity of the polymerization livingness. ^1H NMR spectra of the RAFT polymerization kinetics and the final copolymers are needed in conjunction with SEC data.
- The degree of polymerization is different for the same copolymers. This may have an impact on the LCST behavior of the copolymers. The authors should provide data to support.
- In a recent study (Touve et al., ACS Cent. Sci. 2018, 4, 5, 543–547), it has been reported that electron beam can cause structural damage to polymers, changing chemistry of the copolymers. This can result in the formation of polymeric particles that are different than the targeting one. The authors should investigate this phenomenon and provide data for controlled electron dose study. A much lower electron dose ($0.25 \text{ e}^-/\text{\AA}^2 \text{ s}$) than this work ($0.8 \text{ e}^-/\text{\AA}^2 \text{ s}$) was selected by Touve et al. to minimize the damage. It might give a clue why the phase transitions being observed in this study are partially reversible. For instance, in Figure 3, VT-LCTEM of PEG-b-PDEGMA in water using $0.8 \text{ e}^-/\text{\AA}^2 \text{ s}$ electron dose is given. A similar experiment with much lower electron dose is also given in Figure S19. The particles formed with the lower dose seems to be more uniform and consistent. Similar with PDEGMA in water as shown in Figure 2 and Figure S20.
- The authors should illustrate the partial reversibility of the LCST behavior (e.g. observed in Figure 2,3,4) in Figure 1.
- The morphology observed for PDEGMA homopolymer upon eating seems to be significantly different than what proposed in Figure 1. In Figure S16, authors mention that a different morphology is observed for PEG-b-PDEGMA in water upon repeating the experiment. While some assumptions are given (the role of liquid thickness and potential concentration gradients), additional data to support the assumptions are needed. Can also electron dose play a role? Can it be possible

that some polymeric particles other than the desired ones are formed? Overall, morphology of the homopolymeric and copolymeric particles seems to vary and they are not consistent.

- A recent published work (Scheutz et al., Matter 2021, 4, 722-736) addresses LC-TEM study of thermoresponsive polymers. A comparison between the mentioned paper and this study is needed to highlight the novel contributions of this work.

- Insufficient explanation is given in the main text regarding Figures S7, S15, S19 and S20 and what the authors illustrate. Similar with Figure S10. Additionally, no explanation is given in the main text for Figure S12. The formation of dense polymeric particles being observed (Figure S20) also lacks sufficient explanation.

- Explanation with respect to image processing by MATLAB lacks in the manuscript.

- Structure of the manuscript seems to be flawed and more organized presentation of the study can be helpful.

Reviewer #2 (Remarks to the Author):

The manuscript lacks to cite recent advances in visualization of thermoresponsive nanomaterials via high speed atomic force microscopy, which has been conducted by Dr. Suzuki group in Japan.(e.g., see ACS Omega, pp10836–10842(2018), Angewandte Chemie International Edition, pp8809-8813(2019)). Therefore, the reviewer cannot accept the statement that the novelty of this manuscript is high enough to publish in Nature communications. Additionally, the Suzuki group has already reported the real time analysis of uptake of protein into thermoresponsive nanomaterials imaged by the same method. Thus, future perspective written in this manuscript in conclusion is not novel.

Reviewer #3 (Remarks to the Author):

This manuscript presents thermoresponsive polymer phase transitions in homo, diblock, and triblock copolymers using variable temperature liquid cell TEM (VT-LCTEM) and a variety of supporting techniques. The thermoresponsive block in all cases is based on PDGEMA. Overall, this is a very interesting manuscript that goes to the heart of thermal transitions in block polymers and assemblies. The detailed studies are noteworthy, and the advanced techniques used in the

experimentation are critical to our understanding of the nature of these transitions, which are not understood. I suggest publication after addressing some of the issues I present below:

In Figure 1, the transition in curvature is not necessarily clear in terms of accurately representing the changes in chain conformation at each condition. The authors may want to represent the PDGEMA block as being solvated/extended to the extent of the PEG at low T, and then collapsed similar to the PHEMA in the high temperature case. Here it looks like the chain conformation of the middle block is the same as the end blocks and has little dependence on T. Similarly, the micelle representation in Figure 3 is misleading. This is not a small-molecule micelle – the outside block should be extended and the inside block should be collapsed.

In Figure S2, why does the homopolymer size apparently decrease below the original/aqueous size in the BuOH solution? Is this additive selective towards one of the blocks, or is this simply within error? Along the same lines, in the description of the diblock polymer, the authors state (p. 6) that cooling did not completely redissolve the polymer. They cite diffusion and stickiness to SiNx, but what could be the role of solvent selectivity (beyond being an *OH trap) in these experiments? Finally, for the diblock case, why did the authors switch to 15% IPA? This changes selectivity, and overall dynamics and thermodynamics.

On p. 8, the authors state that they could not measure ionize the diblock micelles using MALDI after exposure, and they attribute this to the polymer being too large. I do not agree with the explanation. Matrix choice is important here and the likely issue.

On p. 8, what is the time scale for small assemblies transforming into larger structures?

In Figure 5(a), is there evidence for spherical micelle formation at low temperature? I think the choice of fit is important, and the expression used should be included in the ESI. I was not really sure what a Lorentzian peak form factor is. Do you mean a spherical form factor accounting for a Lorentzian distribution of particles? If so, you should be able to get some width of the distribution of sizes that can be compared with VT-LCTEM or LS results. In addition, there should be core-shell-shell form factors that can be used for structures like in Figure 6 on the far right. What features would one expect to obtain from the density distributions that would give rise to the form factors?

In Figure S16, concentration gradients are evoked. Can the authors vary concentration to affect this?

Does the confinement of having a 50 nm thick sample cell affect these results at all?

Judd A. and Marjorie Weinberg
College of Arts and Sciences

Robert R. McCormick
School of Engineering & Applied Science

Feinberg School of Medicine

Jacob and Rosaline Cohn Professor
Department of Chemistry
Department of Materials Science & Engineering
Department of Biomedical Engineering
Department of Pharmacology

2145 Sheridan Road
Evanston, Illinois 60208-3113
Phone 858-373-7448
nathan.gianneschi@northwestern.edu

Reviewer #1 (Remarks to the Author)

This is an interesting study to visualize stimulus-responsive behavior of thermo-responsive polymers by LC-TEM. Overall, there are two major concerns related to the chemistry of the homopolymers and copolymers, and TEM imaging. The authors should address these critical comments:

- The amount of initiator (AIBN/ACVA) seems to be very high (up to 10%) for a controlled/living radical polymerization (RAFT). Usually, a less than 0.1mol% is used for living/controlled radical polymerization. Using high quantity of initiator would decrease the livingness of the polymerization and increase polydispersity. The authors should show data indicate the rational behind this system choice.

We thank the reviewer for pointing out our high initiator use. While we were able to achieve polymerization for PDEGMA under an initiator loading of 1% (such that CTA: AIBN was 100: 1), we saw no conversion for the polymerizations we conducted when we employed initiator loadings of less than 5% (such that macro-CTA: ACVA was 50: 1) for the diblock and triblock copolymers. Though using high initiator loadings can affect the control over a polymerization, there is no indication in the literature that high polydispersities can negatively impact the resultant assemblies (Blanazs, A.; Ryan, A. J.; Armes, S. P. *Macromolecules* **2012**, 45 (12), 5099-5107). Moreover, we believe our initiator loading is acceptable because there are numerous examples in the literature of RAFT polymerizations using similar or even higher initiator loadings (Jones, E.; Semsarilar, M.; Wyman, P.; Boerakker, M.; Armes, S. *Polymer Chemistry* **2016**, 7 (4), 851-859, Warren, N. J.; Mykhaylyk, O. O.; Mahmood, D.; Ryan, A. J.; Armes, S. P. *Journal of the American Chemical Society* **2014**, 136 (3), 1023-1033, Blanazs, A.; Ryan, A. J.; Armes, S. P. *Macromolecules* **2012**, 45 (12), 5099-5107). Given the uniformity of the assemblies we observed for the triblock in particular (Figure 4b-c), we are not concerned with our use of 10% initiator. Moreover, the low polydispersities we observed for all PDEGMA-based polymers (<1.25) indicates that our polymerization design is acceptable and controlled. To make the quality of our polymers clear, we have added the polydispersities to the SI in Figure S24.

- The kinetics of the RAFT polymerizations and the corresponding polydispersity should also be shown. This can demonstrate the linearity of the polymerization livingness. ¹H NMR spectra of the RAFT polymerization kinetics and the final copolymers are needed in conjunction with SEC data.

We appreciate the reviewer's comments and have amended our SI to include kinetics studies on all three polymerizations. As requested, we have measured the kinetics via ^1H NMR and SEC-MALS. We have included polydispersity indices for the final polymers in each case. For all polymers, the polydispersity is below 1.2, indicating control over the polymerization.

· The degree of polymerization is different for the same copolymers. This may have an impact on the LCST behavior of the copolymers. The authors should provide data to support.

We thank the reviewer for their thorough review of the polymers we studied. When the reviewer states that the degree of polymerization is different for the "same copolymers," we assume the reviewer means that the block length of a given block (PEG, PDEGMA, PHPMA) is different for different polymers. In the literature, it is well established that PDEGMA has a largely molecular weight independent LCST (Zhang, Q.; Weber, C.; Schubert, U. S.; Hoogenboom, R. *Materials Horizons* **2017**, 4 (2), 109-116). Moreover, our VT-DLS studies showed no indication that the LCST was suppressed or noticeably altered with variation in PDEGMA degree of polymerization (**Figures 2b, 3b, 4c, S2**). As the PEG block remains the same length in both the diblock and triblock copolymers, we assume the reviewer is only concerned with the PDEGMA block. We hope the reviewer is satisfied that the LCST behavior is not largely impacted for different copolymers.

· In a recent study (Touve et al., *ACS Cent. Sci.* 2018, 4, 5, 543–547), it has been reported that electron beam can cause structural damage to polymers, changing chemistry of the copolymers. This can result in the formation of polymeric particles that are different than the targeting one. The authors should investigate this phenomenon and provide data for controlled electron dose study. A much lower electron dose ($0.25 \text{ e}^-/\text{\AA}^2 \text{ s}$) than this work ($0.8 \text{ e}^-/\text{\AA}^2 \text{ s}$) was selected by Touve et al. to minimize the damage. It might give a clue why the phase transitions being observed in this study are partially reversible. For instance, in Figure 3, VT-LCTEM of PEG-b-PDEGMA in water using $0.8 \text{ e}^-/\text{\AA}^2 \text{ s}$ electron dose is given. A similar experiment with much lower electron dose is also given in Figure S19. The particles formed with the lower dose seems to be more uniform and consistent. Similar with PDEGMA in water as shown in Figure 2 and Figure S20.

Electron beam damage is highly sample dependent, and some polymers may be damaged more than others under electron irradiation (Gibson, W.; Patterson, J. P. *Macromolecules* **2021**, 54 (11), 4986-4996). We point the reviewer to Figure 2c of our manuscript, where we conducted an exhaustive electron dose study for the thermoresponsive polymer utilized in our work. Though Touve et al. used a lower flux, their study required more caution as the authors had not conducted *post-mortem* analysis to confirm sample survival under different imaging conditions, whereas we utilized MALDI-IMS in this work. Moreover, in the Touve et al. work, the sample was continuously irradiated when using such a low flux of $0.25 \text{ e}^-/\text{\AA}^2 \text{ s}$, whereas we employed stroboscopic imaging, allowing us to access a comparatively low cumulative flux ($<10 \text{ e}^-/\text{\AA}^2$). As we were able to show polymer survival under the dose conditions we employed, we believe using a flux of $0.8 \text{ e}^-/\text{\AA}^2 \text{ s}$ is acceptable and enables us to gain superior signal-to-noise while still keeping the sample intact. Moreover, at lower fluxes, the particles are clearly more difficult to distinguish and specifically in Figure S19, the particles are not able to be distinguished until liquid-thinning occurred after the sample was held at temperature overnight. Though particle uniformity can be seen in the *post-mortem* dry state, we are more interested here in observing the solution-state morphology of the polymer. Likewise, the experiment in Figure S20 is not the ideal VT-LCTEM experiment as once again the sample was held overnight at temperature, and with the cell dewetting, we were unable to gain any insight into the **solvated** morphology.

· The authors should illustrate the partial reversibility of the LCST behavior (e.g. observed in Figure 2,3,4) in Figure 1.

In response to this comment, we now indicate the possibility of partial reversibility in the figure caption: “Thermally-triggered transformations in an LCST-type polymer for a homopolymer undergoing precipitation(left), a diblock copolymer undergoing assembly (center), and a triblock copolymer undergoing a morphological transformation (right). Note that these transformations may show partial reversibility.”

· The morphology observed for PDEGMA homopolymer upon heating seems to be significantly different than what proposed in Figure 1. In Figure S16, authors mention that a different morphology is observed for PEG-b-PDEGMA in water upon repeating the experiment. While some assumptions are given (the role of liquid thickness and potential concentration gradients), additional data to support the assumptions are needed. Can also electron dose play a role? Can it be possible that some polymeric particles other than the desired ones are formed? Overall, morphology of the homopolymeric and copolymeric particles seems to vary and they are not consistent.

We point here to Figure S14. In this figure, the diblock copolymer was studied under the same exact flux and concentration conditions over several liquid-cell experiments. Despite maintaining the same conditions, a slightly different morphology is observed upon heating in each case. These results seem to indicate that the flux is not the only or even the major contributing factor affecting the sample morphology. Rather, as it is far more difficult to control liquid thickness across experiments, we believe liquid thickness and associated concentration gradients are the main driving forces for the formation of different morphologies. As LCST transitions are driven by polymer-solvent interactions, this is an unsurprising result that LCTEM allowed us to uniquely probe.

· A recent published work (Scheutz et al., Matter 2021, 4, 722-736) addresses LC-TEM study of thermoresponsive polymers. A comparison between the mentioned paper and this study is needed to highlight the novel contributions of this work.

We thank the reviewer for pointing out the connection between this study and the work in Scheutz et al. In that work, the authors utilized VT-LCTEM to initiate thermal PISA whereas in this paper, we aimed to visualize the thermoresponsive behavior of a class of LCST polymers. Our work is the first to utilize VT-LCTEM to probe the stimuli-responsiveness of LCST materials and Scheutz et al. aimed to conduct a polymerization *in situ* under thermal conditions. Finally, this manuscript leverages SAXS analysis, which is lacking from the manuscript by Scheutz et al.

· Insufficient explanation is given in the main text regarding Figures S7, S15, S19 and S20 and what the authors illustrate. Similar with Figure S10. Additionally, no explanation is given in the main text for Figure S12. The formation of dense polymeric particles being observed (Figure S20) also lacks sufficient explanation.

We thank the reviewer for pointing out the insufficient explanation provided for these supplementary figures. We felt it was important to include as many LCTEM datasets as possible to make it clear for readers what type of phenomena may be observed in the *in situ* study of these and related soft materials. We believe providing further explanation in the main text may distract readers from the main findings of our study, but we agree that further explanation is indeed needed. Accordingly, we have added the following passages to the SI:

“Description of LCTEM Experiments with Alcohol Cosolvents

Given the persistence of the cloud point in alcohol mixtures and the ability of alcohols to scavenge destructive •OH radicals, we also attempted to visualize assembly formation in the presence of IPA. In the bulk, PEG-b-PDEGMA assembly formation upon heating occurs with up to 30% IPA, but assembly formation under these solvent conditions was suppressed in the liquid-cell, likely due to concentration gradients in the thin liquid layer (**Figure S22**). However, with 15% IPA, assembly formation in the PEG-b-PDEGMA diblock copolymer was found to occur, though somewhat nonuniformly due to heterogeneities in the IPA distribution in the liquid-cell (**Figure S7**). We also attempted to visualize assembly formation in

5% t-BuOH. Though assembly formation still occurs in the bulk for up to 10% t-BuOH, in over five VT-LCTEM experiments using only 5% t-BuOH, we saw weak evidence of assembly formation in only one experiment (**Figure S22**). Given the inherently thin liquid layer required for LCTEM analysis, the failure of the LCST transition in the presence of t-BuOH may be due the existence of high local concentrations of t-BuOH that prevent effective polymer-water interactions critical to the cloud point transition.

“Description of PEG-b-PDMAEMA System

With the PEG-b-PDEGMA diblock system, we demonstrate the first example of a thermoresponsive diblock copolymer assembly being directly observed in solution upon heating to temperature. Notably, our approach for studying thermoresponsive diblock copolymers is generalizable, and we were also able to visualize the elevated temperature morphology of PEG-b-poly(2-(dimethylamino)ethyl methacrylate) (PDMAEMA), which likewise contains a thermoresponsive PDMAEMA block (**Scheme S4, Figure S12**). Note that we were not able to observe nanostructure-assembly in any system we studied with polymer concentrations less than 10 mg mL^{-1} , as we have found that a high local polymer concentration is required in the diffusion-limited, liquid-cell environment (**Figure S13**).”

“Polymer-Solvent Interactions During VT-LCTEM Experiments

In several VT-LCTEM experiments, we observed liquid thinning or exclusion from the liquid-cell window upon heating (**Figure S15, S19, S20**). As the LCST transition is a manifestation of polymer-solvent interactions, changes in liquid thickness under LCTEM conditions are not entirely surprising. Notably, in two experiments with PEG-b-PDEGMA, liquid thinning upon heating was observed upon the phase separation of the PDEGMA block (**Figure S12, S19**). Likewise, the PDEGMA homopolymer exhibited a clear phase separation upon overnight heating, featuring a dense polymeric phase and a water phase (**Figure S20**).”

- Explanation with respect to image processing by MATLAB lacks in the manuscript.

We point the reviewer to the SI of our manuscript where we list out the functions utilized in MATLAB in the captions of Figures S9 and S14. For clarity, we have added some discussion on this processing to section III of the SI. Additionally, we have added some discussion on the processing performed in FIJI to obtain the data in Figure 4 to section III of the SI as shown below:

“III. Image Processing Details

To probe the morphological transformation of the triblock further, we performed image processing using the software FIJI. First, we cropped a region of interest in a fixed area for each timepoint (**Figure 4d**). We binned the cropped images (2×2 average), applied a gaussian filter ($\sigma = 1$), thresholded each cropped region of interest, and subtracted features less than 5 pixels^2 from each image (**Figure 4e**). The intermediate contrast of the corona was better captured using a Lookup Table (LUT), whereby each pixel value range was assigned a corresponding red, green, and blue value. Using a 6-shade LUT enabled colorization and thus allowed for a clearer visual distinction between multiple features with different intensity values (**Figure 4f**).

For the MATLAB image processing showed in **Figure S9** and **S14**, we cropped each image to only feature the electron transparent SiN_x membrane, and we enhanced the images with `imadjust`, `histeq`, and `adapthisteq`, which are functions available with the image processing toolkit in MATLAB.”

- Structure of the manuscript seems to be flawed and more organized presentation of the study can be helpful.

We hope that the incorporation of edits as outlined above, and in response to other reviewer comments has improved the structure of the manuscript.

Reviewer #2 (Remarks to the Author):

The manuscript lacks to cite recent advances in visualization of thermoresponsive nanomaterials via high speed atomic force microscopy, which has been conducted by Dr. Suzuki group in Japan.(e.g., see ACS Omega, pp10836–10842(2018), Angewandte Chemie International Edition, pp8809-8813(2019)). Therefore, the reviewer cannot accept the statement that the novelty of this manuscript is high enough to publish in Nature communications. Additionally, the Suzuki group has already reported the real time analysis of uptake of protein into thermoresponsive nanomaterials imaged by the same method. Thus, future perspective written in this manuscript in conclusion is not novel.

We thank the reviewer for bringing this critical work to our attention. As with LCTEM, the imaging probe used in HS-AFM studies can potentially alter the sample. Accordingly, we believe that our work, the first to utilize VT-LCTEM to study these types of thermoresponsive nanomaterials, is indeed novel and can further aid in the understanding gained from studies like that of the Suzuki group. The strengths and limitations of novel *in situ* nanoscopy techniques, including LCTEM and liquid-AFM, can and should certainly be debated. However, we believe that expanding the toolkit of *in situ* methods available for chemistry and material science research is extremely valuable, and here LCTEM has demonstrated its particular utility for directly observing phase transitions in soft matter solvated nanostructures, which has not been possible to date using other *in situ* nanoscopy methods. Nonetheless, given the importance of the Suzuki group's work in the realm of HS-AFM, we have added a citation to this work in our manuscript, as shown below. We also note that the work done by the Suzuki group on thermoresponsive PNIPAM microspheres is distinct from our own work as we are concerned with block copolymer assembly dynamics. "Thermoresponsive polymers are used in numerous technological applications, including biomedicine, insulator materials, and tissue engineering.^{1,2,3,4,5}"

Reviewer #3 (Remarks to the Author):

This manuscript presents thermoresponsive polymer phase transitions in homo, diblock, and triblock copolymers using variable temperature liquid cell TEM (VT-LCTEM) and a variety of supporting techniques. The thermoresponsive block in all cases is based on PDGEMA. Overall, this is a very interesting manuscript that goes to the heart of thermal transitions in block polymers and assemblies. The detailed studies are noteworthy, and the advanced techniques used in the experimentation are critical to our understanding of the nature of these transitions, which are not understood. I suggest publication after addressing some of the issues I present below:

In Figure 1, the transition in curvature is not necessarily clear in terms of accurately representing the changes in chain conformation at each condition. The authors may want to represent the PDGEMA block as being solvated/extended to the extent of the PEG at low T, and then collapsed similar to the PHEMA in the high temperature case. Here it looks like the chain conformation of the middle block is the same as the end blocks and has little dependence on T. Similarly, the micelle representation in Figure 3 is misleading. This is not a small-molecule micelle – the outside block should be extended and the inside block should be collapsed.

We thank the reviewer for pointing out the issues with Figures 1 and 3. We have corrected the representation of the transition in Figure 1 and have removed the problematic portion of the representation of the micelle in Figure 3.

In Figure S2, why does the homopolymer size apparently decrease below the original/aqueous size in the BuOH solution? Is this additive selective towards one of the blocks, or is this simply within error? Along the same lines, in the description of the diblock polymer, the authors state (p. 6) that cooling did not

completely redissolve the polymer. They cite diffusion and stickiness to SiNx, but what could be the role of solvent selectivity (beyond being an *OH trap) in these experiments? Finally, for the diblock case, why did the authors switch to 15% IPA? This changes selectivity, and overall dynamics and thermodynamics.

We thank the reviewer for noticing this subtle detail in Figure S2. The DLS used for these studies is not sensitive enough to distinguish nanomaterials below <10 nm so the reviewer is correct that this result is within error. We chose to use 15% IPA because we observed the persistence of the cloud point of PDEGMA with even 30% IPA in the bulk but were unable to visualize a response *in situ* with such a high IPA concentration (Figure S22). Though admittedly a high IPA loading, our previous work showed the effectiveness of IPA at mitigating polymer damage at similarly high concentrations (Korpanty, J.; Parent, L. R.; Gianneschi, N. C. *Nano Letters* **2021**, *21* (2), 1141-1149.), and we sought to evaluate whether our previous results would apply to a thermoresponsive polymer. We agree with the reviewer, however, that adding such a high concentration of IPA would affect the overall dynamics of the LCST transition given the distinct formation of vesicles under these solvent conditions (Figure 3e, S7b).

On p. 8, the authors state that they could not measure ionize the diblock micelles using MALDI after exposure, and they attribute this to the polymer being too large. I do not agree with the explanation. Matrix choice is important here and the likely issue.

We agree with the reviewer's statement that matrix choice is important for successful MALDI analysis. However, for MALDI-IMS specifically, the spot size is limited to 50 microns, and this results in a weaker ionization source, preventing larger polymers from being easily ionized. Using the typical MALDI setup, we were able to ionize the diblock copolymer, but could not do so under the MALDI-IMS spot size constraints. To prevent confusion for readers, we did not include this MALDI data in the SI of the paper but have included it below for the reviewer.

On p. 8, what is the time scale for small assemblies transforming into larger structures?

We refer the reviewer to Figure 4 where we have indicated the timepoint for each image. The sample was heated for a total of 50 minutes and cooled for 2 hours.

In Figure 5(a), is there evidence for spherical micelle formation at low temperature? I think the choice of fit is important, and the expression used should be included in the ESI. I was not really sure what a Lorentzian peak form factor is. Do you mean a spherical form factor accounting for a Lorentzian distribution of particles? If so, you should be able to get some width of the distribution of sizes that can be compared with VT-LCTEM or LS results. In addition, there should be core-shell-shell form factors that can be used for structures like in Figure 6 on the far right. What features would one expect to obtain from the density distributions that would give rise to the form factors?

The reviewer brings up a good question that spurred us to reevaluate our choice of models for fitting the variable temperature SAXS data. SAXS data obtained for the triblock at 30 °C (below the LCST of PDEGMA) was background subtracted and fit to a core-shell-shell sphere model, comprising a PHPMA core, a PDEGMA shell, and a PEG corona. At 30 °C, the PHPMA block is expected to aggregate into the micelle core, while the hydrophilic PDEGMA and PEG blocks are expected to remain hydrated and extended into the aqueous solution. The thicknesses of the PDEGMA and PEG shells were estimated as 1.5 and 4.2 nm, respectively, and fixed during data fitting. Thicknesses were assumed to be twice the R_g of each block, where R_g was calculated using the known degrees of polymerization and statistical segment lengths and assuming scaling for a good solvent. To account for the polydispersity of the micelle size, a log-normal distribution with a dispersity of 0.3 was used for the core, shell, and corona. The fitting parameters were the scattering length density of each block, the core radius, and an arbitrary intensity pre-factor which was used to account for the non-exact intensity units. The PHPMA core and PDEGMA shell SLDs were nearly identical to the theoretical values, supporting the validity of the fit. The PEG corona SLD was close to that of water, reflecting the highly hydrated nature of the shell. The PHPMA shell radius was found to be 18 nm, resulting in an overall micelle radius of approximately 23 nm, consistent with the values obtained by TEM and DLS. All fitting parameters and a description of the fitting process were included in the SI.

Based on the reasonable success we had with fitting our experimental data to a core-shell-shell model at 30 °C, we attempted to perform a similar fit to the data acquired at 60 °C (above the LCST of PDEGMA). At 60 °C, both the PHPMA and PDEGMA blocks are expected to be collapsed into the micelle core, while the PEG block remains hydrophilic and hydrated in the micelle corona. Although, PHPMA forms the micelle core at 30 °C, we expect that there will be a large entropic penalty for packing the PHPMA domain in the core above the LCST of PDEGMA due to the relatively long length of the PHPMA block as compared to the PDEGMA block. Since 60 °C is above both the LCST of PDEGMA and the T_g of PHPMA, we hypothesize that the micelle core rearranges to accommodate PDEGMA in the core center to relieve the entropic penalty by placing the shorter PDEGMA block on the concave side of the core-shell interface. The resulting micelles thus contain a phase segregated PHPMA/PDEGMA core surrounded by a PEG corona. Our hypothesis is supported by the persistence of the scattering features observed at 60 °C even after slowly cooling back to 30 °C, suggesting that the PDEGMA core is trapped because PHPMA vitrification precedes crossing the LCST during cooling. In other words, the T_g of PHPMA is higher than the LCST of PDEGMA such that the core becomes kinetically trapped before PDEGMA can swell and induce core rearrangement. During fitting, the thickness of the PEG corona was fixed as 4.2 nm based on $2 \cdot R_g$. Fitting parameters were scattering length density of each block, the core radius, the inner shell radius, and an arbitrary intensity pre-factor which was used to account for the non-exact intensity units. We were unable to obtain a satisfactory fit to the experimental data using reasonable fitting parameters. To verify whether the poor fit was due to an incorrect hypothesis that placed PDEGMA in the micelle core, we also fit to a core-shell-shell sphere model using similar fitting parameters but assuming that the micelle core was PHPMA and the inner shell was PDEGMA. Again, we failed to obtain a satisfactory fit using reasonable fitting parameters. In particular, we have struggled to model the broad peak around 0.2 nm^{-1} using a standard core-shell-shell sphere model. Based on these struggles, we believe that the true structure of the micelles above the LCST of PDEGMA is significantly more complex than a core-shell-shell sphere model assumes. Instead, we believe that the micelle core more resembles a multicomponent micelle, where discrete spherical domains of PDEGMA are embedded within a PHPMA core.

Therefore, we decided to fit the experimental data to the sum of a spherical and a broad peak form factor, where the broad peak form factor overlays a Lorentzian peak onto a power law decay. The spherical form factor is intended to capture the overall shape of the micelle, while the broad peak form factor is intended to capture the multiphase nature of the micelle core, namely the characteristic spacing between scattering inhomogeneities in the core. Although the broad peak model is unable to provide precise information regarding how the PDEGMA and PHPMA domains are partitioned within the micelle core, it can provide

the necessary domain spacings. Similar sums of spherical and broad peak form factors have previously been reported for other soft spherical nanoparticles with multicompartmental cores, e.g., lipid nanoparticles loaded with mRNA (Sebastiani, F. et al. *ACS Nano* **2021**, *15*, 6709-6722 and Valldeperas, M. et al. *Soft Matter* **2019**, *10*, 2178-2189). Fitting parameters for our model include: sphere SLD, sphere radius, an arbitrary intensity pre-factor, Porod exponent, Porod scale, Lorentzian scale, Lorentzian exponent, Lorentzian screening length, and peak position. The fitting process yielded a sphere radius of 95 nm and a spacing of approximately 30 nm between inhomogeneities in the micelle core, which are both reasonably consistent with the TEM data. A summary of the fitting parameters and an explanation of the fitting method has been added to the SI.

Based on these comments, the initial SAXS discussion in the manuscript was replaced with the passage below:

“With the initial hypothesis that the triblock forms spherical micelles with a PHPMA core, a PDEGMA shell, and a PEG corona below the LCST of PDEGMA, we fit the experimental scattering data acquired at 30 °C to a core-shell-shell sphere form factor. The fitted curve captured the experimental data reasonably well for intermediate and high q , although significant deviation was observed in the low q region, which we attribute to the presence of loose micellar aggregates (**Figure 5b**). The fit yielded a core radius of 18 nm, a PDEGMA inner shell thickness of 1.5 nm, and a PEG outer shell thickness of 4.2 nm. These sizes are reasonably consistent with the DLS and TEM measurements, supporting the robustness of our fitting process and demonstrating the potential for using SAXS as a complementary technique to LCTEM (Table S1).

Heating the sample to 60 °C resulted in the formation of a broad peak in the high q region of the SAXS data, which supports the LCTEM observation that a morphological change occurs above the PDEGMA LCST. Based on the low degree of polymerization of the PDEGMA block compared to the PHPMA block, we speculated that the micelle core would restructure above the LCST to accommodate PDEGMA in the core interior and relieve the large entropic penalty for arranging the shorter block on the convex side of the core-shell interface. The LCST of PDEGMA is higher than the T_g of PHPMA, allowing the mobile PHPMA core to accommodate the collapsed PDEGMA domain at 60 °C. Furthermore, the persistence of the broad peak even after cooling back to 30 °C suggests that the PDEGMA domain is trapped within a vitrified PHPMA matrix and is unable to rehydrate below the LCST. Using this hypothesized PDEGMA core, PHPMA inner shell, and PEG outer shell configuration as our initial model, we attempted to fit the 60 °C SAXS data to the same core-shell-shell sphere form factor that we successfully used for the 30 °C data. However, a satisfactory fit was unobtainable using physically relevant fitting parameters. To confirm that the fitting challenge was unrelated to our choice of initial model, we also attempted to fit the data to the core-shell-shell form factor assuming a PHPMA core, a PDEGMA inner shell, and a PEG outer shell. Again, a satisfactory fit was unobtainable. Therefore, we speculated that the true micelle structure was more complex than the concentric spheres assumed by the core-shell-shell model. Instead, we hypothesized that the PHPMA and PDEGMA domains may be microphase segregated within the micelle core. To test this hypothesis, we then fit the experimental scattering data obtained at 60 °C to the sum of a spherical and broad peak form factor, where the spherical form factor captures the overall micelle shape and the broad peak form factor captures the phase segregated core (**Figure 5c**). This choice of scattering model yielded an acceptable fit to the experimental data with a micelle radius of 95 nm and a 30 nm spacing between scattering inhomogeneities within the micelle core. The fitted values for the micelle radius correspond well to those measured by LCTEM and VT-DLS, while the 30 nm length scale is reasonably consistent with the size of the high contrast region observed in the micelle interior by LCTEM.”

Additionally, the following was added to the SI:

Data was fit to well-defined form factor models using the SASView software (<http://www.sasview.org/>). With the assumption that the triblock assembles into a spherical micelle with a PHPMA core, a PDEGMA inner shell, and a PEG outer shell below the LCST of PDEGMA, data obtained at 30 °C was fit to a core-shell-shell sphere form factor:

$$P(q) = \frac{3V_{core}}{qR_{core}} (\rho_{core} - \rho_{inner}) J_{core}(qR_{core}) + \frac{3V_{inner}}{qR_{inner}} (\rho_{inner} - \rho_{outer}) J_{inner}(qR_{inner}) + \frac{3V_{outer}}{qR_{outer}} (\rho_{solvent} - \rho_{outer}) J_{outer}(qR_{outer})$$

where R is the radius of the core or thickness of the shells, V is the volume of the core or shell ($V = 4\pi R^3/3$), ρ is the scattering length density, and J is a first order Bessel function of the form

$$J = \frac{\sin(x) - x\cos(x)}{x^2}$$

During the fitting process, the inner and outer shell thicknesses were fixed after estimating based on known degrees of polymerization, statistical segment lengths, and assuming scaling for a good solvent. Solvent SLD, the incoherent background intensity, and the log-normal size distributions of the core, inner shell, and outer shell were also fixed. All other parameters were fit using SASView. Extracted fit parameters as well as fixed parameters are included below (**Table S1**). Bolded values indicate that these parameters were extracted during the fitting process. The fit deviates from the experimental data at low q , suggesting the presence of loosely formed aggregates.

Table S1. Extracted Fitting Parameters for at 30 °C SAXS Trace

Parameter	Value
Intensity Pre-factor	0.03
Background	0.01
SLD _{PHPMA}	22.9E-6 A⁻²
Core Radius	18 nm
SLD _{PDEGMA}	25.9E-6 A⁻²
Inner Shell Thickness	1.5 nm
SLD _{PEG}	25.8E-6 A⁻²
Outer Shell Thickness	4.2 nm
SLD _{H2O}	26.1E-6 A ⁻²
Core Size Distribution	0.3
Shell Size Distribution	0.3
Corona Size Distribution	0.3

We attempted to fit the scattering data acquired above the LCST at 60 °C to the same core-shell-shell sphere form factor with the assumption that the hydrated PDEGMA block at 30 °C becomes hydrophobic and collapses into the micelle core upon crossing its LCST. The short length of the PDEGMA block as compared to the PHPMA block paired with the persistence of the scattering features observed at 60 °C upon cooling back to 30 °C suggests that PDEGMA resides within the interior of the micelle core and is surrounded by a PHPMA shell. The PEG outer shell remains hydrophilic and extended into the aqueous solvent. Using these assumptions, we attempted to fit the 60 °C scattering data to the core-shell-shell sphere model where PDEGMA is the micelle core, PHPMA is the inner shell, and PEG is the outer shell. Again, the thickness of the outer shell, the SLD of the solvent, the incoherent background intensity, and the log-normal size distributions were fixed. The fitting parameters were thus the core radius, the core SLD, the inner shell thickness, the inner shell radius, the outer shell SLD, and an arbitrary pre-factor to account for the non-exact units of intensity. However, despite our success with fitting the data at 30 °C, we were unable to attain a satisfactory fit using reasonable fitting parameters. To ensure that the poor convergence was not a consequence of an incorrect assumption of micelle structure, we repeated the fitting process using an initial guess of a PHPMA core, a PDEGMA inner shell, and a PEG outer shell, yet we were still unable to obtain a satisfactory fit. These fitting challenges suggested that the true micelle structure was likely more complex than the series of concentric spheres assumed by the core-shell-shell form factor. Instead, we believe that the PDEGMA domains may be discrete islands within a PHPMA matrix within the micelle core. To reflect this hypothesis, we thus fit the data to the sum of a spherical form factor and a broad peak form factor, where the broad peak form factor overlaid a Lorentzian peak onto a power

law decay. Such a form factor has previously been used to model scattering data from for other soft spherical nanoparticles with multicompartmental cores, e.g., lipid nanoparticles loaded with mRNA.^{7, 8} The spherical form factor was intended to capture the overall shape of the micelle, while the broad peak form factor was intended to describe the compartmentalized core. The form factor is provided below:

$$P(q) = [3V\Delta\rho \cdot \frac{\sin(qR) - qR\cos(qR)}{qR^3}]^2 + \frac{A}{q^n} + \frac{B}{1 + (|q - q_0|\xi)^m}$$

where V is the sphere volume, R is the sphere radius, A is the Porod law scale factor, n is the Porod exponent, B is the Lorentzian scale factor, q_0 is the broad peak position, ξ is the screening length, and m is the q scaling exponent. Fitting parameters for our model include: sphere SLD, sphere radius, an arbitrary intensity pre-factor, Porod exponent, Porod scale, Lorentzian scale, Lorentzian exponent, Lorentzian screening length, and peak position. The fit yielded a spherical micelle with a radius of 95 nm with a core that had microphase separated scattering inhomogeneities with a spacing of 30 nm. The extracted fitting parameters are provided below (**Table S2**). Fitted parameters are bolded.

Table S2. Extracted Fitting Parameters for at 60 °C SAXS Trace

Parameter	Value
Intensity Pre-factor	0.03
Background	0.01
SLD _{PHPMA}	22.9E-6 A⁻²
Core Radius	18 nm
SLD _{PDEGMA}	25.9E-6 A⁻²
Inner Shell Thickness	1.5 nm
SLD _{PEG}	25.8E-6 A⁻²
Outer Shell Thickness	4.2 nm
SLD _{H2O}	26.1E-6 A ⁻²
Core Size Distribution	0.3
Shell Size Distribution	0.3
Corona Size Distribution	0.3
Parameter	Value
Intensity Pre-factor	0.1
Background	0.005
SLD _{sphere}	25.9E-6 A⁻²
Sphere Radius	95 nm
SLD _{H2O}	26.1E-6 A ⁻²
Core Size Distribution	0.3
Shell Size Distribution	0.3
Corona Size Distribution	0.3
Porod Scale	4.62E-8
Porod Exponent	4
Lorentz Scale	4.32
Lorentz Screening Length	5.5 nm
Peak Position	0.17 nm⁻¹
Lorentz Exponent	4.0

In Figure S16, concentration gradients are evoked. Can the authors vary concentration to affect this?

Given the slightly different assembly behavior observed in the experiments shown in Figure S16, which were conducted under identical imaging and concentration conditions, we believe it would be difficult to have control over concentration gradients in the liquid-cell. We believe these gradients and variations in

liquid thickness across LCTEM experiments are a consequence of the dropcasting procedure, in which one dropcasts sub-microliter quantities of sample before placing the top chip. As the dropcasted volume is so low and evaporates quickly, it is difficult to have precise control. More control could be potentially gained by flowing the sample into the holder, but for polymeric materials, there is a risk of the sample becoming lodged in the tubing and consequently being diluted. Note, with respect to varying the concentration, we learned that assemblies could not be visualized *in situ* under more dilute conditions of 5 mg mL⁻¹ (Figure S13).

Does the confinement of having a 50 nm thick sample cell affect these results at all?

We do believe that the unavoidable confinement of the LCTEM setup may affect the results; this may explain why some LCTEM experiments showed slightly different polymeric morphologies. Though the liquid-cell chips are fabricated with 50 nm spacers, SiN_x bulging under vacuum leads to thicker liquid layers. Accordingly, the liquid is thinnest in the corners of the liquid-cell, which is why we chose to image in these regions. We note that even in the corners, the liquid is still likely thicker than the 50 nm spacer size. While thicker liquid layers may enable the sample to exhibit behavior more representative of bulk conditions, for LCTEM of low Z polymeric materials, thicker liquid layers often prevent the sample from being observable under non-damaging low flux conditions. Despite these potential complications, the LCTEM confinement did not prevent the thermoresponsive behavior of the polymers to be studied and variations from one experiment to another were minimal.

REVIEWERS' COMMENTS

Reviewer #3 (Remarks to the Author):

This manuscript was a revision to a previous version. I have carefully read the revised manuscript and the responses to all three reviewers. I appreciate the effort with which the authors addressed the important issues from the reviewers. I have some additional comments below:

(1) In the authors' response to R1's third comment, I agree with the authors that molecular weight differences here should have minimal effects on the LCST.

(2) In the authors' response to R1's eighth comment, the authors attempt to address the experiments in the presence of alcohols (IPA, t-BuOH) and the effect on assembly. They state that in the "bulk," the systems can withstand higher alcohol content, but this is dramatically reduced in the liquid TEM cell. The explanation they give is that this effect is due to concentration fluctuations, and I would like the authors explain this a little more. From where do the fluctuations come? Would this be in the thickness direction, presumably from interactions with the liquid cell windows?

(3) With regard to R2, I do see the novelty in this study and consider this a significant deviation from high-speed AFM. The additional references in response to this comment definitely help frame the context of the current study in the context of previous work.

(4) With regard to R3, I appreciate the detailed description for the SAXS fitting routine and consider the authors to have addressed this concern well.

Judd A. and Marjorie Weinberg
College of Arts and Sciences

Robert R. McCormick
School of Engineering & Applied Science

Feinberg School of Medicine

Jacob and Rosaline Cohn Professor
Department of Chemistry
Department of Materials Science & Engineering
Department of Biomedical Engineering
Department of Pharmacology

2145 Sheridan Road
Evanston, Illinois 60208-3113
Phone 858-373-7448
nathan.gianneschi@northwestern.edu

Reviewer #3 (Remarks to the Author):

This manuscript was a revision to a previous version. I have carefully read the revised manuscript and the responses to all three reviewers. I appreciate the effort with which the authors addressed the important issues from the reviewers. I have some additional comments below:

(1) In the authors' response to R1's third comment, I agree with the authors that molecular weight differences here should have minimal effects on the LCST.

We thank the reviewer for their agreement.

(2) In the authors' response to R1's eighth comment, the authors attempt to address the experiments in the presence of alcohols (IPA, t-BuOH) and the effect on assembly. They state that in the "bulk," the systems can withstand higher alcohol content, but this is dramatically reduced in the liquid TEM cell. The explanation they give is that this effect is due to concentration fluctuations, and I would like the authors explain this a little more. From where do the fluctuations come? Would this be in the thickness direction, presumably from interactions with the liquid cell windows?

We thank the reviewer for their inquiry. The presence of concentration gradients in LCTEM is a well-established phenomenon that we and others have observed in previous works (Woehl, T. J.; Abellan, P., *Journal of Microscopy* **2017**, 265 (2), 135-147. and Parent, L. R.; Bakalis, E.; Ramírez-Hernández, A.; Kammeyer, J. K.; Park, C.; de Pablo, J.; Zerbetto, F.; Patterson, J. P.; Gianneschi, N. C., *Journal of the American Chemical Society* **2017**, 139 (47), 17140-17151.). The reviewer is correct in believing that window-sample interactions are the culprit. Accordingly, we have added the following text to the supplementary information:

"Given the inherently thin liquid layer required for LCTEM analysis, the failure of the LCST transition in the presence of t-BuOH may be due the existence of t-BuOH concentration gradients, with areas of high and low local t-BuOH concentrations, that prevent effective polymer-water interactions critical to the cloud point transition. The presence of concentration gradients in liquid-cell experiments is likely a manifestation of the high surface area of the liquid-cell windows compared to bulk conditions. Additionally, slight variations in the solution concentration during loading the low sample volumes

required for LCTEM could also play a role in the formation of concentration gradients.”

(3) With regard to R2, I do see the novelty in this study and consider this a significant deviation from high-speed AFM. The additional references in response to this comment definitely help frame the context of the current study in the context of previous work.

We agree with the reviewer and thank them for acknowledging the novelty of our work.

(4) With regard to R3, I appreciate the detailed description for the SAXS fitting routine and consider the authors to have addressed this concern well.

We thank the reviewer for their comment.